# FIGHTING FIRE WITH FIRE: CONTRASTIVE DEBIASING WITHOUT BIAS-FREE DATA VIA GENERATIVE BIAS-TRANSFORMATION

## ABSTRACT

Despite their remarkable ability to generalize with over-capacity networks, deep neural networks often abuse bias instead of using the actual task-related information for discriminative tasks. Since such shortcuts are only effective within the collected dataset, the resulting biased model underperforms on real-world inputs. To counteract the influence of bias, existing methods either exploit auxiliary information which is rarely obtainable in practice or sift handful bias-free samples to emphasize them for debiasing. However, such presumptions are not always guaranteed, and the existing methods could break down due to the unmet presumptions. In this paper, we propose Contrastive Debiasing via Generative Bias-transformation (CDvG) which is capable of operating without exploiting bias labels and bias-free samples explicitly. Motivated by our observation that not only discriminative models but also image translation models tend to focus on the easy-to-learn bias, CDvG employs a image translation model to transform the bias to another mode of bias while preserving task-relevant information. Through contrastive learning, we set transformed biased views against another, learning bias-invariant representations. Especially, as the bias has a stronger correlation or is easier to perceive compared to the signal, the translation model is more likely to be a bias translation model, resulting in better debiasing effect. Experimental results demonstrate that CDvG outperforms the state-of-the-arts, especially when bias-free samples are extremely scarce.

## 1 INTRODUCTION

Recent advances in deep learning have showcased that deep neural networks (DNN) are capable of reaching state-of-the-art performance in various fields of machine learning, such as computer vision (He et al., 2015), natural language processing (Brown et al., 2020), reinforcement learning (Mnih et al., 2016) and more. However, it is also known that the over-parameterized nature of DNNs not only exposes them to general overfitting but also renders them susceptible to biases present in collected datasets (Torralba & Efros, 2011) which are detrimental to the generalizability. In supervised learning, neural networks tend to prefer shortcut solutions based on biases rather than real signal (Zhu et al., 2017b; Li et al., 2018). Since spurious correlations do not provide task-related information, DNNs that use these biases will ultimately fail on future data. For instance, a classifier trained to identify *car racing* images using a dataset dominated by track will exploit the track road information. However, the classifier will fail to exhibit the same performance on images of off-road rallies. To this end, debiasing is imperative in utilizing DNNs for real-world applications.

A tautological solution to the bias problem is to construct a bias-free dataset from the start. However, curating a dataset devoid of all bias is extremely costly at best, and generally infeasible. A more practical attempt at neutralizing dataset bias is to fortify a dataset with explicit supervision with regards to the bias (Kim et al., 2019a; Sagawa et al., 2019). However, additional expenditure of human labor in procuring such information cannot be avoided, which renders the option less appealing.

In most cases where such explicit supervision for bias is absent, the following two lines of works are recently proposed. One line of works mitigates the influence of bias by leveraging the bias type (e.g. texture) (Bahng et al., 2020; Geirhos et al., 2019; Wang et al., 2019; Hong & Yang, 2021)

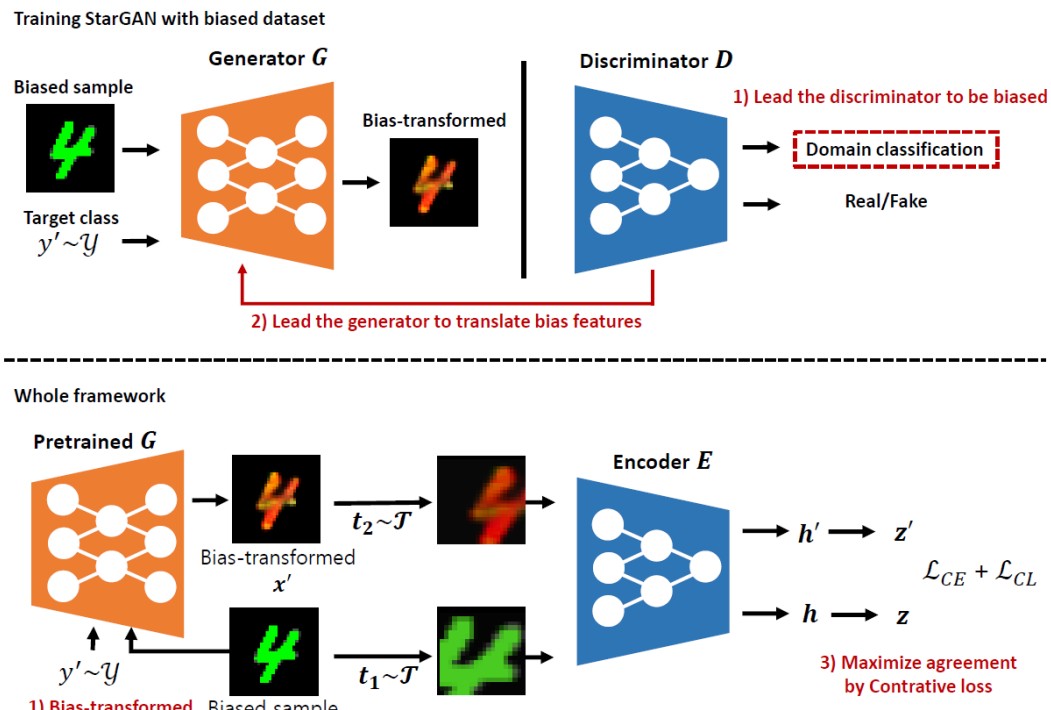

Figure 1: Illustration of our Contrastive Debiasing via Generative Bias-transformation (CDvG).

to design bias-oriented auxiliary models or to augment texture-perturbed samples. However, such prior knowledge of bias is by no means guaranteed, and even with such information, designing bias-oriented architectures is not always straightforward. Instead, another line of works leverages an empirical prior knowledge that malignant biases are usually learned faster than task-relevant features (Li & Vasconcelos, 2019; Nam et al., 2020; Lee et al., 2021; Kim et al., 2021). However, these methods tend to break down in regimes where bias-free samples are extremely scarce or absent (See Section 4.1).

To tackle these shortcomings, we develop a general debiasing method capable of operating even in regimes devoid of bias data, thus not reliant on any presumptions regarding the biases. In this paper, we propose **C**ontrastive **D**ebiasing **v**ia **G**enerative Bias-transformation (CDvG), which contrasts the bias modes within the dataset against each other to attenuate the influence of bias while effectively learning the task-relevant information contained in all samples. Through preliminary experiments, we find that image-to-image translation models favor learning malignant biases over task-relevant signals, as discriminative models are known to do (See Section 3.2). Motivated by the observation, we train a biased image translation model without bias labels that learns the bias distribution over the signal, obtaining capabilities to transform the bias feature of a given input to another bias. Using the learned bias-translation model, we synthesize novel views with altered bias features. Fighting fire with fire, we pit one bias type against another via contrastive learning. By maximizing agreement between the views with different biases, the model is encouraged to learn bias-invariant representations. Unlike existing methods, CDvG does not require explicit supervision, domain knowledge, or other meta-knowledge - the existence of bias-free samples.

Our contributions are three-fold:

- We experimentally observe that certain image translation models are also prone to consider conspicuous but incidental information such as background or texture first rather than task-related information, as discriminative models do (Sec. 3).

- We propose **C**ontrastive **D**ebiasing **v**ia **G**enerative Bias-transformation (CDvG) with generative view transformation that transforms bias factors of input images (Sec. 4). CDvG does not presume the existence of bias labels, bias type information, or even the existence

of bias-free samples which is more malignant. Also, CDvG can be integrated with other debiasing methods in a plug-and-play manner.

- With experiments on synthetic and real-world datasets, we demonstrate that CDvG outperforms baselines in a variety of bias situations (Sec. 5). We show that CDvG is particularly effective compared to existing methods when bias-free samples are extremely scarce or absent.

## 2 RELATED WORKS

In real-world datasets across diverse domains, there exist various kinds of incidental biases strongly but spuriously correlated with the task-related information. However, when the bias factors are more noticeable and easier to learn than the task-related signals, DNNs tend to lean on such biases (Nam et al., 2020), causing failures in generalization. To counter this effect, several lines of works were developed.

**Explicit bias supervision** can be used to screen or mitigate the influence of bias. Kim et al. (2019a) uses known explicit bias supervision to train an auxiliary network that helps in reducing the influence of bias. Sagawa et al. (2019) use the bias supervision to group data samples for grouped distributionally robust optimization. Kim et al. (2019b) address the problem by minimizing the mutual information not to learn biased instances by using bias supervision. Hong & Yang (2021) propose bias-contrastive loss and bias-balanced regression that encourages the model to pull together the samples in the same class with different bias features, with balancing target-bias distribution.

**Domain knowledge of biases.** When acquiring bias supervision is impractical, we can leverage domain-specific knowledge about bias type. For example, it was shown that ImageNet-trained classifiers exploit texture information in the image rather than information contained in the object of interest (.Gatys et al., 2017; Brendel & Bethge, 2019). Utilizing this fact, Geirhos et al. (2019) construct an augmented dataset by applying various textures to the images for texture debiasing. On the other hand, the following works exploit an auxiliary model which is carefully designed to capture biases over signals. Wang et al. (2019) trains a texture-debiased classifier by projecting classifier representations to a subspace orthogonal to the previously learned biased space. Bahng et al. (2020) lead the model representation to be statistically independent to the biased one produced from the biased auxiliary model using Hilbert-Schmidt independence criterion. Hong & Yang (2021) further propose a soft version of bias-contrastive loss for the case where the type of bias is available. However, these methods are grounded on obtaining such knowledge - which is often costly, or even impossible. In addition, designing a bias-oriented auxiliary model may not be so intuitive depending on the type of bias.

**Without domain knowledge.** To combat bias without any side information, Li & Vasconcelos (2019) introduced an alternating minimization scheme between bias identification and sample reweighting. Using the fact that biases are usually learned faster than salient task-relevant features, Nam et al. (2020) train an auxiliary model slightly ahead in terms of training iterations using generalized cross entropy (Zhang & Sabuncu, 2018) to absorb *fast-learned* bias, and assign more weight to the bias-free samples for the debiased follower. Seo et al. (2022) cluster the training samples into the $K$ groups based on the latent feature of the Empirical Risk Minimization(ERM) model and reweight the loss of each sample in a batch according to the assigned groups. Lee et al. (2021) proposed feature-level data augmentation that disentangles bias features by using a biased auxiliary model obtained by following Nam et al. (2020) and swaps latent bias features within the mini-batch. Another augmentation-based debiasing method, BiaSwap (Kim et al., 2021), employs task-related features in abundant bias-aligned samples by synthesizing a new image that takes the bias-irrelevant core features from the biased sample and the bias attribute from the bias-free sample. However, all these methods presume that bias-free samples do exist in sufficient quantities and can be distinguished, which cannot always be guaranteed.

**Contrastive learning** is a self-supervised learning method proven to learn representations substantially beneficial to numerous downstream tasks, achieving state-of-the-art performance (Chen et al., 2020a;b; Grill et al., 2020; Chen & He, 2020; Khosla et al., 2020). A representative work (Chen et al., 2020a) defines the contrastive prediction task in two steps: a) augmenting two views from the same image by strong data augmentations (Bachman et al., 2019; Hénaff et al., 2019; Krizhevsky et al., 2017) and b) maximizing agreement between the augmented views on the latent space by

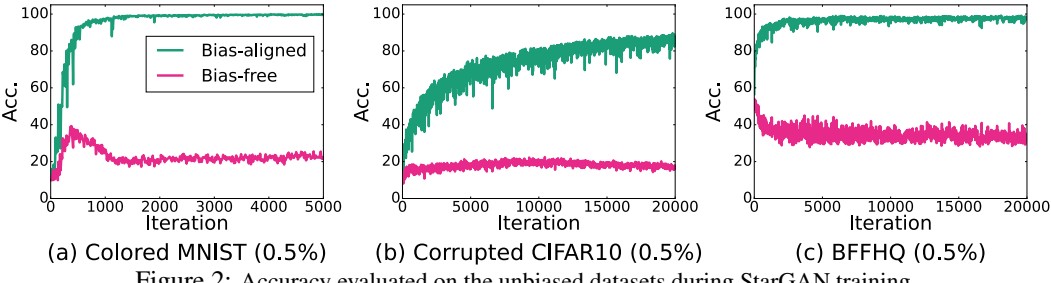

Figure 2: Accuracy evaluated on the unbiased datasets during StarGAN training.

employing the contrastive loss (van den Oord et al., 2018) to capture core features of the image. In the process of maximizing the agreement between the augmented views, the encoder discards the deviating features between the views and learns transformation-invariant representations. For example, using color distortion methods such as *color jitter* and *gray scale* in the augmentation step encourages the encoder to learn color-invariant representations.

## 3   IMAGE TRANSLATION MODELS FOR GENERATING BIAS-FREE SAMPLES

Previous approaches to addressing the bias problem is to either deploy manpower or algorithms to salvage a sufficient amount of bias-free samples from the contaminated dataset. However, under circumstances where bias-free samples are almost nonexistent, there are not enough bias-free samples to begin with. Instead of salvaging, we first opt to synthesize bias-free samples using image translation models. In other words, we aim to translate bias-aligned samples to bias-free samples by changing a given bias factor to another.

In this section, we first define our setup in Section 3.1 and describe our observations of how translation models behave on biased datasets, which is a key part of our algorithm, in Section 3.2.

### 3.1   SETUP

To formally define our target task, we introduce the following random variables: input image $X$, target label $Y$, signal $S$, bias $B$ and other attributes $O$ comprising input $X = (S, B, O)$. Here, $B$ is a feature unrelated to $Y$, that is, $Y$ and $B$ are independent given $S$. We further define the random variable for bias label $Y_B$ which is hidden.

Ideally, the image classification model parameterized by $\theta$ predicts the label based on signal as $P_\theta(Y|X) = P_\theta(Y|B,S) = P_\theta(Y|S)$. However, when the training data consists of highly but spuriously correlated bias and target, i.e., $H(Y^{tr}|B^{tr}) \approx 0$, predicting $Y$ based on $B$ is also one of the possible solutions that can be deemed effective in the training phase. In this paper, when the sample $x = (s, b, o)$ consist of a correlated signal and a bias (i.e. high $P_{tr}(b|s)$), $x$ is called *bias-aligned* and the opposite case is called *bias-free*. We tackle the case where $B$ is easier to perceive than $S$, so that the spurious correlation between $B$ and $Y$ is *malignant* in that the model preferentially takes $B$ as a clue to predict $Y$ over $S$ (Nam et al., 2020). This is obviously an unintended consequence and impairs generalizability due to the discrepancy of the bias-target correlation $Y|B$ between the training and test phases.

### 3.2   BEHAVIOR OF IMAGE TRANSLATION MODELS UNDER BIAS

We first examine whether image translation models are capable of generating bias-free samples. Since discriminative models are known to be susceptible to bias, it is not far-fetched to first enquire whether generative models also carry the same frailty. For instance, one might intuitively suspect that the family of Generative Adversarial Networks are susceptible to bias, as their composition by nature involves a discriminator, which is bias-pregnable. To this end, we investigate the case of a particular member of the GAN family, namely, image translation models.

Image-to-image translation methods (Isola et al., 2017; Zhu et al., 2017a; Choi et al., 2017) render an image $x$ from a source domain $y$ to a target domain $y'$. Ideally, they find out the representative characteristics of the target domain $y'$ and combine them with the input image. However, when the dataset is biased, the translation model interacting with the biased domain classifier is also prone to

translate biases rather than task-related domain features to satisfy the domain classifier (See Figure 1). For example, CycleGAN (Zhu et al., 2017a), a representative milestone of image translation, presented a number of typical failure modes that when the source domain is an apple and the target domain is an orange, a transformed image is not the orange counterpart of the input, but an apple with the color and texture of an orange. This implies that the model perceives color and texture rather than shape as the representative traits for the target domain, even without using a biased dataset. With this in consideration, it is plausible to speculate that this phenomenon would be exacerbated when handling highly biased datasets. To verify this, we examine the behavior of StarGAN (Choi et al., 2017) on biased datasets.

### 3.2.1 DISCRIMINATORS OF TRANSLATION MODELS ARE ALSO PRONE TO BE BIASED.

StarGAN introduces an auxiliary domain classifier $D_{cls}$ on top of discriminator $D$ to enable translation between multiple domains. The domain classifier $D_{cls}$, trained on the real image with domain labels, learns to classify images with representative traits of the domains by optimizing the domain classification loss of real images $\mathcal{L}^r_{cls} = \mathbb{E}_{(x,y)\sim\mathcal{D}}[-\log D_{cls}(y|x)]$. However, on the biased dataset, we observed that $D_{cls}$ has absorbed bias attributes as representative traits of the domains during a training phase. To quantitatively evaluate whether $D_{cls}$ is biased, we measure the classification loss on the unbiased dataset with $D_{cls}$ while training StarGAN on the biased dataset Colored MNIST, Corrupted CIFAR10, and BFFHQ (Figure 4) which have color bias, texture bias, and gender bias, respectively. Figure 2 shows the classification losses of bias-aligned and bias-free samples, respectively. We observed that the accuracy of bias-aligned samples increases to near 100%, however, the accuracy of bias-free samples is low depending on how malignant the bias of each data set is. Therefore, we conclude that $D_{cls}$ utilizes the biases rather than the task-related features.

### 3.2.2 A BIASED DOMAIN CLASSIFIER INDUCES A BIASED TRANSLATION GENERATOR.

By optimizing the domain classification loss $\mathcal{L}^f_{cls} = \mathbb{E}_{(x,y)\sim\mathcal{D},y'\sim\mathcal{Y}}[-\log D_{cls}(y'|G(x,y'))]$ of translated fake images $G(x,y')$, biased $D_{cls}$ induces the generator $G$ to translate the bias attribute of image $x$ into the other bias correlated to the randomly sampled target domain $y'$ rather than the task-relevant signal. As a result, $G$ becomes a bias-translator. This phenomenon becomes more noticeable as the bias is malignant for $D_{cls}$ - that is, the more scarce the bias-free samples are or the easier the bias is to perceive. To quantitatively evaluate whether the translation model truly favor learning biases over task-relevant signals, we measure the classification loss of translated images $x' = G(x,y')$ with the bias classifier $C_B$ and the signal classifier $C_S$ while training StarGAN on the biased dataset Colored MNIST and BFFHQ. As alternatives of oracles, the classifiers $C_B$ and $C_S$ are trained with the bias label $y_B$ and the true class label $y$, respectively, on the unbiased Colored MNIST and FFHQ. The bias and signal loss are defined as $\mathbb{E}_{(x,y)\sim\mathcal{D},y'\sim Cat(|\mathcal{Y}|)}[\mathcal{L}_{CE}(C_B(x'),y')]$ and $\mathbb{E}_{(x,y)\sim\mathcal{D},y'\sim Cat(|\mathcal{Y}|)}[\mathcal{L}_{CE}(C_S(x'),y')]$, respectively.

As expected, Figure 3 shows that the bias is more favorable to translation model. For Colored MNIST, the bias loss quickly declines to zero, but the signal loss increases rapidly since $G$ concentrates on the color biases rather than digits. For BFFHQ, both signal and bias loss are decreased due to the less malignant bias (see vanilla performance in Table 1), however, $G$ still places more emphasis on the bias.

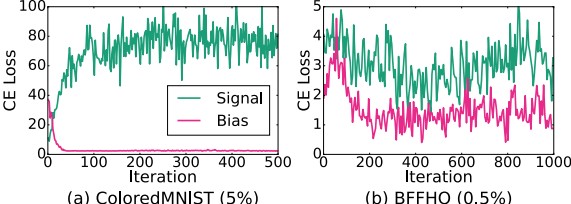

Figure 3: Classification loss evaluated on the bias-translated images during StarGAN training.

Next, we also present the qualitative results of the biased translator $G$ in Figure 4. The leftmost column of each subfigure is of the original $x$ and the other columns are bias-translated results of each class. Interestingly, we found that the translated results retain their contents to some extent and color, texture, background (e.g. rock wall for climbing, water for fishing), and gender characteristics (e.g. makeup for female, beard for male), which are the respective bias features of the datasets, are altered. It is worth mentioning that we can take the advantage of the image translation model to identify bias attributes by distinguishing unintentionally translated features from the translated images. For example, when StarGAN has trained on CelebA with blond hair attribute, it not only turns a male photo into blond hair, but also applies makeup which reveals gender bias.

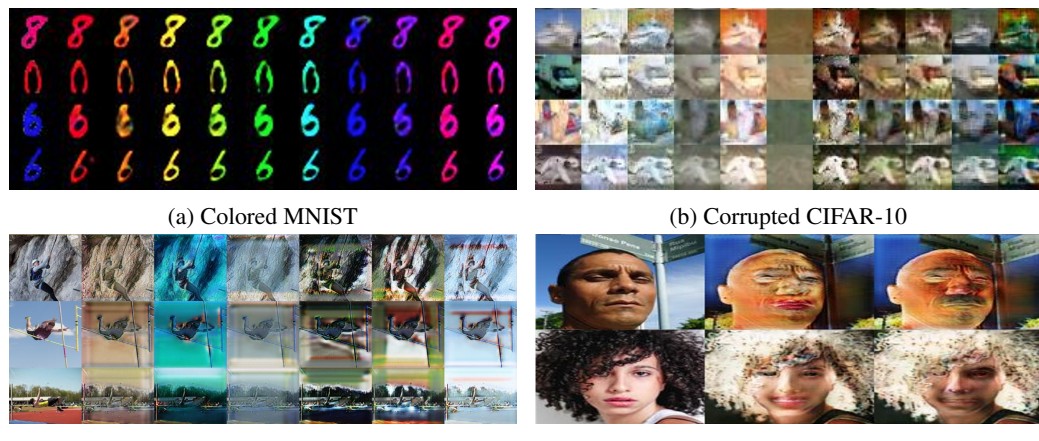

(a) Colored MNIST                                    (b) Corrupted CIFAR-10

(c) BAR                                              (d) BFFHQ

Figure 4: Bias-transformed images by our generative bias-transformation on synthetic and real-world biased datasets. The leftmost column of each subfigure contains the original images and each column is the transformed images of each target domain. The resulting images show the translated bias attributes such as color (Colored MNIST), texture (Corrupted CIFAR-10), background (BAR), and gender (BFFHQ) ,respectively. For details about the datasets, we refer the reader Section A in Supplementary.

Although generated images of BFFHQ in Figure 4d look quite different from the real images to humans, the core contents are still discernible while the gender-specific features emerge as pink lips and eye makeup in the second column and thick eyebrow and beard for the third column. Also, note that the biased classifier judges that the bias-transformed images are well transformed into the target label with high confidence. These observations verify that there is a chance that image translation models trained on biased datasets can be employed to synthesize bias-free samples for debiasing.

## 4 METHOD

In this section, we empirically show that the previous unsupervised debiasing methods break down when the presumption about the availability of bias-free samples does not hold in Section 4.1. Acknowledging this limitation, we propose a new debiasing method called CDvG in Section 4.2 leveraging our empirical findings (Section 3.2) that image translation models are prone to be biased.

### 4.1 LEARNING WITHOUT BIAS-FREE SAMPLES

Although recent debiasing methods (Bahng et al., 2020; Hong & Yang, 2021; Nam et al., 2020; Lee et al., 2021; Kim et al., 2021) work as intended when they can exploit the bias-free samples, they neglect the case where such bias-free samples are extremely scarce or absent. Thus, in regimes where a presumption that enough number of bias-free samples exist does not hold, their behaviors are practically unknown, and are likely to break down. More specifically, in Table 1, we empirically demonstrate that previous methods fail to effectively debias the model (showing low performance, almost comparable to Vanilla) when the proportion of bias-free samples is 0%. Also, we further analyse the sample reweighting scheme (Nam et al., 2020; Lee et al., 2021) whose key idea is to balance bias-free and bias-aligned samples. To this end, we compare the total weights of bias-free samples and bias-aligned samples as the ratio of bias-free samples decreases. We observe that, as the ratio of bias-free samples decreases, the debiasing effect decreases because the bias-free samples are not sufficiently emphasized by the baselines. The detailed descriptions and results are deferred to Section C due to the space constraint. Based on this observation, we consider a more general task of learning-under-bias that handles the situations where there are no to only a few bias-free samples as well as the cases where such samples are sufficiently provided.

### 4.2 CONTRASTIVE DEBIASING WITHOUT BIAS-FREE DATA VIA BIAS-TRANSFORMED VIEWS

Motivated by the findings of Section 3.2, we propose a novel debiasing method called Contrastive Debiasing via Generative Bias-transformation (CDvG) that employs the biased translation model

---

**Algorithm 1:** Contrastive Debiasing via Generative Bias-transformation (CDvG)

---

**Input:** Encoder $E$, Projection head $H$, Classifier $C$, Biased generator $G$, Augmentation family $\mathcal{T}$
**Data:** Training set $\mathcal{D} = \{(x, y)\} \subset \mathcal{X} \times \mathcal{Y}$

1  **for** minibatch $\{(x_k, y_k)\}_{k=1}^{N}$ **do**
2      # *Generate bias-transformed view*
3      $y'_k \sim Categorical(|\mathcal{Y}|) ; \quad x'_k = G(x_k, y'_k)$
4      # *Augmentation operator*
5      $t_1 \sim \mathcal{T}, t_2 \sim \mathcal{T} ; \quad \tilde{x}_k, \tilde{x}'_k = t_1(x_k), t_2(x'_k)$
6      **Update** $E, C, H$ to minimize
7      $\sum_k (\mathcal{L}_{CE}(C(E(\tilde{x}_k)), y_k) + \mathcal{L}_{CE}(C(E(\tilde{x}'_k)), y_k)) + \lambda * \mathcal{L}_{CL}(E, H)$
8  **end for**

---

to transform an image to have different biases corresponding to other classes and integrate it with the contrastive learning framework. By comparing the diverse bias-translated views generated with the transformation function, the contrastive loss encourages to learn bias-invariant representations without distinguishing bias-free data explicitly. The whole process is provided in the following paragraphs and summarized in Algorithm 1.

First, we train StarGAN (Choi et al., 2017) on a training dataset, which is biased in our problem setting, to obtain the bias-transformation generator $G$. We observe in Section 3.2 that the image translation model learns biases as the main characteristics of the domain rather than task-related features. Although other multi-class domain translation models can be used, we adopt StarGAN, the representative image-to-image translation model. It is worth noting that we do not need bias-free samples at hand in this process. With $G$, we translate an input image $x$ with a target label $y'$ to obtain a bias-transformed view $x' = G(x, y')$, where $y'$ is uniformly sampled for every iteration (Step 1 in Figure 1, Line 3 in Alg. 1) to generate views with diverse biases. Please note that bias-translation with $G$ is applied only for training, not at the test time.

After the bias-transformation step, we additionally apply the random augmentation operators $t_1$ and $t_2$ to the original and the bias-transformed images respectively as $\tilde{x} = t_1(x)$, $\tilde{x}' = t_2(x')$, where $t_1$ and $t_2$ are sampled from the same augmentation family $\mathcal{T}$ (Step 2 in Figure 1, Line 5 in Alg. 1). According to Chen et al. (Chen et al., 2020a), strong augmentations are essential to the performance of the contrastive learning framework as they prevent the encoder from easily finding a clue about the fact that two views come from the same image. By following (Chen et al., 2020a), $\mathcal{T}$ is composed of the following sequential augmentations: random resized cropping and random horizontal flipping. Please note that we did not adopt color distortion methods because most biases are related to color.

Finally, the resulting learning objective is given by the combination of the cross entropy loss and the contrastive loss (Step 3 in Figure 1, Line 7 in Alg. 1). First, the encoder $E$ and the following classifier $C$ are optimized to minimize the cross entropy loss $L_{CE}$ which is applied to both $\tilde{x}$ and $\tilde{x}'$:

$$\min_{E,C} \mathcal{L}_{CE}(C(E(\tilde{x})), y) + \mathcal{L}_{CE}(C(E(\tilde{x}')), y).$$

Also, we train the encoder $E$ and the projection head $H$ to minimize the contrastive loss $\mathcal{L}_{CL}$ to tie the original view $\tilde{x}$ and the bias-transformed view $\tilde{x}'$ as $\mathcal{L}_{CL}(E, H) = \sum_{k=1}^{N} \ell(2k-1, 2k) + \ell(2k, 2k-1)$. The loss $\ell(i, j)$ for a positive pair $(i, j)$ is defined as

$$\ell(i, j) = -\log \frac{\exp(sim_{i,j}/\tau)}{\sum_{k=1}^{2N} \mathbb{1}_{k \neq i} \exp(sim_{i,k}/\tau)},$$

Where $sim_{i,j} = z_i^\top z_j / (||z_i||||z_j||)$ is the cosine similarity, and $z_i = H(E(\tilde{x}_i))$ is a projected representation of $\tilde{x}_i$ with the base encoder $E$ followed by the projection head $H$. By tying the original view $\tilde{x}$ and the bias-transformed view $\tilde{x}'$ as a positive pair, the encoder attempts to attenuate biases while emphasizing the true signals shared by the views via maximizing the mutual information between their latent representations. Please note that we adopt 1.0 for the coefficient of contrastive loss $\lambda$ by default.

Table 1: The average and the standard deviation of accuracy over 3 runs. ✓ indicates that the model exploits auxiliary information of the bias and ✗ does not.

| Dataset | Ratio(%) | Vanilla ✗ | ReBias ✓ | LfF ✗ | EnD ✗ | BiaSwap ✗ | BPA ✗ | CDvG ✗ | CDvG+LfF ✗ |
|---|---|---|---|---|---|---|---|---|---|
| Colored MNIST | 0.0 | $12.53_{\pm0.92}$ | $14.64_{\pm0.50}$ | $13.16_{\pm1.87}$ | $11.65_{\pm0.61}$ | - | $10.08_{\pm0.04}$ | $\underline{96.16}_{\pm0.48}$ | $\mathbf{96.48}_{\pm0.19}$ |
| | 0.5 | $39.12_{\pm0.91}$ | $70.47_{\pm1.84}$ | $66.54_{\pm3.80}$ | $62.13_{\pm3.96}$ | 85.76 | $54.52_{\pm3.39}$ | $\underline{95.73}_{\pm0.14}$ | $\mathbf{96.20}_{\pm0.12}$ |
| | 1.0 | $56.02_{\pm1.81}$ | $87.40_{\pm0.78}$ | $79.83_{\pm2.23}$ | $75.49_{\pm0.21}$ | 83.74 | $72.63_{\pm0.27}$ | $\underline{96.13}_{\pm0.48}$ | $\mathbf{96.45}_{\pm0.19}$ |
| | 2.0 | $69.32_{\pm0.22}$ | $92.91_{\pm0.15}$ | $82.66_{\pm0.39}$ | $80.08_{\pm0.45}$ | 85.29 | $78.52_{\pm0.59}$ | $\underline{96.90}_{\pm0.22}$ | $\mathbf{96.97}_{\pm0.17}$ |
| | 5.0 | $83.93_{\pm0.89}$ | $\underline{96.96}_{\pm0.04}$ | $83.30_{\pm1.23}$ | $85.00_{\pm1.17}$ | 90.85 | $85.30_{\pm0.93}$ | $96.73_{\pm0.05}$ | $\mathbf{96.98}_{\pm0.05}$ |
| Corrupted CIFAR-10 | 0.0 | $16.05_{\pm0.13}$ | $21.93_{\pm0.37}$ | $15.88_{\pm0.45}$ | $18.76_{\pm0.88}$ | - | $17.14_{\pm1.54}$ | $\underline{28.44}_{\pm0.21}$ | $\mathbf{29.24}_{\pm0.47}$ |
| | 0.5 | $20.87_{\pm0.34}$ | $22.27_{\pm0.41}$ | $25.58_{\pm0.23}$ | $28.62_{\pm1.74}$ | 29.11 | $25.50_{\pm1.03}$ | $\underline{31.50}_{\pm0.33}$ | $\mathbf{39.07}_{\pm0.27}$ |
| | 1.0 | $24.05_{\pm0.61}$ | $25.72_{\pm0.20}$ | $30.68_{\pm0.50}$ | $32.31_{\pm0.03}$ | 32.54 | $26.86_{\pm0.69}$ | $\underline{33.25}_{\pm0.20}$ | $\mathbf{43.81}_{\pm0.35}$ |
| | 2.0 | $29.47_{\pm0.20}$ | $31.66_{\pm0.43}$ | $\underline{37.96}_{\pm1.09}$ | $36.51_{\pm2.34}$ | 35.25 | $27.47_{\pm1.46}$ | $35.16_{\pm0.23}$ | $\mathbf{47.45}_{\pm0.07}$ |
| | 5.0 | $41.12_{\pm0.16}$ | $43.43_{\pm0.41}$ | $\underline{48.49}_{\pm0.16}$ | $46.41_{\pm0.62}$ | 41.62 | $34.29_{\pm2.20}$ | $42.75_{\pm0.19}$ | $\mathbf{52.31}_{\pm0.13}$ |
| BFFHQ | 0.0 | $37.93_{\pm0.96}$ | $43.47_{\pm0.74}$ | $39.67_{\pm1.00}$ | $38.13_{\pm2.13}$ | - | $48.20_{\pm1.40}$ | $\underline{48.80}_{\pm1.18}$ | $\mathbf{49.60}_{\pm1.18}$ |
| | 0.5 | $52.40_{\pm1.88}$ | $56.80_{\pm1.56}$ | $\underline{58.07}_{\pm0.82}$ | $54.33_{\pm0.92}$ | - | $51.40_{\pm2.98}$ | $54.80_{\pm0.33}$ | $\mathbf{62.20}_{\pm0.45}$ |

## 5 EXPERIMENTS

To validate the effectiveness of CDvG compared to recent debiasing methods, we conduct image classification experiments on standard benchmark datasets for debiasing. We first present the settings for our experiment including datasets and baselines (Section 5.1) then report the comparisons of our method and baselines regarding their debiasing performance (Section 5.2). In addition, we perform ablation studies to demonstrate that each component of CDvG contributes to respective performance improvements (Section 5.3). Also, we provide implementation details in Section B.

### 5.1 EXPERIMENTAL SETTINGS

**Dataset.** We experiment on {Colored MNIST, Corrupted CIFAR-10} and {BFFHQ, ImageNet-9 (IN-9), Waterbirds} which are synthetic datasets injected with synthetic biases and real-world datasets with natural biases, respectively. We provide detailed description of datasets in Section A. By setting the proportion of bias-free samples to {0%, 0.5%, 1%, 2%, 5%}, we evaluate the performance considering the highly biased setting and, moreover, the hardest case where bias-free samples are absent. The datasets with 0% bias-free samples are constructed by excluding bias-free samples from the datasets with the ratio of 0.5%.

**Baselines.** To benchmark the performance of CDvG, we compare CDvG with diverse categories of debiasing methods. We compared with HEX (Wang et al., 2019) and Rebias (Bahng et al., 2020) which leaveraging the domain knowledge about bias and LfF (Nam et al., 2020) which deliberately trains a auxiliary biased classifier to distinguish and utilize bias-free samples. Also, DisEnt (Lee et al., 2021) and BiaSwap (Kim et al., 2021), which are based on the bias-free sample selection scheme of (Nam et al., 2020), try to combine the other biases with the instance. For BiaSwap, we borrow the available performance directly from the papers and remain as blank for the others because the official code of BiaSwap is unavailable. Also, we compared with BPA (Seo et al., 2022) and EIIL (Creager et al., 2021) which are based on ERM model.

### 5.2 RESULTS

We report the results on the standard benchmark datasets to validate the effectiveness of CDvG for debiasing when bias label is not available. We observed that the CDvG outperforms the baselines especially when the bias-free samples are scarce(0.5%) or absent(0%).

we further proposed to combine CDvG and LfF ('CDvG+LfF' in Table 1) to leverage both bias-aligned and bias-free samples which is consistently superior to the baselines with a low standard deviation in Table 1. In more details, in the training process, we trained CDvG with concurrently applying LfF's sample-wise weighting to CE loss for both original images and bias-translated images. Therefore, 'CDvG+LfF' allows us to utilize both bias-aligned and bias-free samples to debias by translating bias-aligned samples into bias-free samples (CDvG) and giving more weights to bias-free samples (LfF). Please note that CDvG can be integrated with other debiasing methods in a plug-and-

Table 2: ImageNet-9 dataset

| Dataset | Test type | Vanilla | RUBi | LfF | Rebias | CDvG+LfF |
|---------|-----------|---------|------|-----|--------|----------|
| IN-9 | Biased | $94.0_{\pm 0.1}$ | $93.9_{\pm 0.2}$ | $91.2_{\pm 0.1}$ | $94.0_{\pm 0.2}$ | $\mathbf{95.2}_{\pm 0.1}$ |
| | Unbiased | $92.7_{\pm 0.2}$ | $92.5_{\pm 0.2}$ | $89.6_{\pm 0.3}$ | $92.7_{\pm 0.2}$ | $\mathbf{94.5}_{\pm 0.1}$ |
| | IN-a | $30.5_{\pm 0.5}$ | $31.0_{\pm 0.2}$ | $29.4_{\pm 0.8}$ | $30.5_{\pm 0.2}$ | $\mathbf{34.6}_{\pm 0.4}$ |

Table 3: Waterbirds dataset. The average of accuracy over 3 runs. We borrow the performance on ResNet-18 and ResNet-50 directly from (Seo et al., 2022) and (Creager et al., 2021), respectively.

| Dataset | Test type | ResNet-18 | | | | ResNet-50 | | |
|---------|-----------|-----------|-----|-----|----------|-----------|------|----------|
| | | Vanilla | LfF | BPA | CDvG+LfF | ERM | EIIL | CDvG+LfF |
| Waterbirds | Unbiased | 84.63 | 85.48 | **87.05** | 86.25 | **97.30** | 96.90 | 91.30 |
| | Worst-group | 62.39 | 68.02 | 71.39 | **74.92** | 60.30 | 78.70 | **84.80** |

play manner. In addition, we evaluate the performance when bias-free samples do not exist (ratio 0%) which is likely to occur in real-world settings but not considered in recent studies.

**Synthetic datasets** In Table 1, we observe that the CDvG preforms well on Colored MNIST and Corrupted CIFAR-10 when the bias-free samples are scare or absent, i.e., when the biases are more malignant. CDvG+LfF, which is an our integrated method, outperforms the baselines on overall ratio with a large margin.

**Real-world datasets** We conducted experiments on BFFHQ, IN-9 and Waterbirds to evaluate the performance on real-world setting. CDvG+LfF shows improved performance with small standard deviations on BFFHQ compared to state-of-the-art. Also shows improved or comparable performance on IN-9 and Waterbirds in Table 2 and Table 3, respectively.

To sum up, our method can handle various types of synthetic and real-world biases and works well on a wide range of bias ratios, especially when bias-free samples are scarce and even absent, compared with state-of-the-art methods Overall consistent results empirically demonstrate that our method debias better than previous approaches that only focus on bias-free samples.

### 5.3 ABLATION STUDY

We study the effectiveness of each component of CDvG. In Table 4, CDvG without bias-transformation generator (CDvG w/o $G$), which is adding augmentation operator $\mathcal{T}$ and contrastive loss to vanilla, shows degradation. It mainly comes from that augmented samples, which are still biased, accelerate the model becomes biased. By adopting our bias-transform generator $G$ only for augmentation (CDvG w/o CL), there is 8.21%, 5.37%, 4.07%, 2.22%, and 1.43% improvements for each ratio, respectively. It shows that our generator successfully augments bias-transformed images which work well for some extent. However, since bias-transform generator $G$ is not always a perfect model that only translate biases, contrastive learning is needed. Our whole framework shows an obvious improvement which means that since bias-transformed images are not perfect, it is essential to induce bias-invariance and capture the true signal shared by the bias-transformed view and original view by contrastive loss. Based on the ablation results, we have shown that our whole framework effectively address the bias problem rather than a single component.

Table 4: Ablation study for bias-transformed view generation (w/o G) and for contrastive loss (w/o CL).

| Dataset | Ratio(%) | Vanilla | CDvG w/o $G$ | CDvG w/o CL | CDvG |
|---------|----------|---------|--------------|-------------|------|
| Corrupted CIFAR-10 | 0.0 | $16.04_{\pm 0.13}$ | $18.29_{\pm 0.90}$ | $24.25_{\pm 0.15}$ | $\mathbf{28.44}_{\pm 0.21}$ |
| | 0.5 | $20.87_{\pm 0.34}$ | $20.75_{\pm 0.21}$ | $26.24_{\pm 0.12}$ | $\mathbf{31.50}_{\pm 0.33}$ |
| | 1.0 | $24.05_{\pm 0.61}$ | $21.37_{\pm 0.29}$ | $28.12_{\pm 0.21}$ | $\mathbf{33.25}_{\pm 0.20}$ |
| | 2.0 | $29.47_{\pm 0.20}$ | $24.50_{\pm 0.34}$ | $31.69_{\pm 0.41}$ | $\mathbf{35.16}_{\pm 0.23}$ |
| | 5.0 | $41.12_{\pm 0.16}$ | $31.13_{\pm 0.56}$ | $42.55_{\pm 0.50}$ | $\mathbf{42.75}_{\pm 0.19}$ |

## 6  CONCLUSION

In this paper, we have proposed Contrastive Debiasing via Generative bias-transformation, a general debiasing method that does not require presumptions about the data such as the existence of bias-free samples or any domain knowledge. Motivated by the observation that not only discriminative models but also image translation models tend to focus on bias, we utilize a translation model as a bias-transformation function to generate diverse biased views. With contrastive learning, we compare the biased views to obtain bias-invariant representations. Our method can be integrated with other debiasing methods in a plug-and-play manner and especially shows good synergy when integrating with models that focus on bias-free samples such as LfF. Our experimental results show that our method outperforms the current state-of-the-arts in both synthetic and real-world biases, especially when bias-free samples are scarce or even absent.

One important future work direction, and a current limitation, is the extension to datasets that bias-free data are the majority. Including CDvG, many recent debiasing approaches are focus on highly biased setting where bias-aligned are the dominant majority. Although such situations would not require debiasing techniques from the start, suggesting a robust method that work in general is an important area of ongoing research.

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
