# OpenReview forum: "Fighting Fire with Fire: Contrastive Debiasing without Bias-free Data via Generative Bias-transformation"
_ICLR.cc/2023/Conference — Submitted to ICLR 2023_

### Official Review · Reviewer_Xnts · 2022-10-17

**Confidence:** 4
**Correctness:** 3
**Technical Novelty And Significance:** 3
**Empirical Novelty And Significance:** 3
**Recommendation:** 5

**Clarity, Quality, Novelty And Reproducibility:**

**Clarity:**
The paper is clearly written except for some sections e.g., Sec 5.3 (Ablation Study) which needs to be revised heavily due to usage of terms such as

‘there is remarkable improvement’/ ‘obvious improvement’ – better just provide a range of gains and remove terms such as remarkable/obvious

‘bias-transform generator G is not always a perfect model’? → what is a ‘perfect G’? Did you mean to say, it may alter signal/core features too?

**Awkward writing:**
‘... address the bias problem not a single component … ‘

The subsection needs to be re-written, but this poorly written paragraph did not affect my score.

**Novelty:** As noted in the paper, the observation that DNNs find it easier to latch onto biases compared to the signal  (at least for the types of datasets studied in this line of work) is a known observation. However, the application to image translation models is novel.

Reproducibility: It lacks supplementary materials (appendix/code), so reproducibility is questionable.


**Strength And Weaknesses:**

**Strengths (S):**

[S1] The approach is neat and sensible. It converts the weakness of DNNs (that image translation models latch onto biases) to a strength (use that to generate diverse biases).

[S2] The results show benefits over existing methods on the datasets tested. However, I have some concerns about the choice of datasets (see weaknesses section).

[S3] The method functions well despite not having bias-free samples in the training set, which is an important result.

**Weaknesses (W):**

I think the experiments leave many open questions about the true generalization abilities of the system:

[W1] The paper claims that it doesn’t require presumptions about the data, but it assumes that the biases are easier to exploit than the signal. What if they are of similar complexities to exploit/learn? Wouldn’t the generative model then alter the core features too and not just the bias?

[W2] ColoredMNIST consists of single source of bias. But how would this method handle multiple types of biases e.g., BiasedMNIST [1]? Does the image translation model uncover all the spurious factors regardless of their ‘ease of exploitation’?

[W3] Waterbirds, BAR, ImageNet-9 all test against foreground-background spurious correlations. Apart from completeness, do they add additional value?

[W4] It is unclear how the method would tackle real world distributional shifts. Examples,  datasets from WILDs benchmark [2] or the different renditions of ImageNet (https://github.com/hendrycks/imagenet-r) could be used, which are not tested here.

[W5] The original submission does not include references or the appendix which is referenced in the main text.

[1] Shrestha, Robik, Kushal Kafle, and Christopher Kanan. "OccamNets: Mitigating Dataset Bias by Favoring Simpler Hypotheses." arXiv preprint arXiv:2204.02426 (2022).

[2]  Koh, Pang Wei, et al. "Wilds: A benchmark of in-the-wild distribution shifts." International Conference on Machine Learning. PMLR, 2021.

[3] Hendrycks, Dan, et al. "The many faces of robustness: A critical analysis of out-of-distribution generalization." Proceedings of the IEEE/CVF International Conference on Computer Vision. 2021.

**Summary Of The Paper:**

The paper uses image translation models to generate diverse biased views of images to debias models. For this, the paper observes that generative models find it easier to latch onto biases instead of the signal, which is consistent with the observation made by previous works for classification models. As such, when asked to translate, they change the bias features, while not altering the core features, thus providing views with different biases.

**Summary Of The Review:**

I think the paper cleverly exploits the bias-prone nature of generative models to debias methods, which is an important contribution.
While I am leaning toward acceptance, I cannot see this as a clear accept mainly due to the experimental limitations.
I would also need confirmation if the authors have fixed Sec 5.3, and included references and the appendix.

---

> ### Author Response · Authors · 2022-11-17
> **Response to Reviewer Xnts**
>
> We sincerely appreciate your time and effort to review our paper. We address the reviewer’s concerns below.
>
> ### [W1. The paper claims that it doesn’t require presumptions about the data, but it assumes that the biases are easier to exploit than the signal. What if they are of similar complexities to exploit/learn? Wouldn’t the generative model then alter the core features too and not just the bias?]
> As we mentioned in Sec. 3.1, we tackle the case where the bias is easier to perceive than the signal, so that the spurious correlation between bias and label is malignant in that the model preferentially takes the bias as a clue to predict the label over signal. Otherwise we do not need a debiasing model.
>
> It is also important to note that in our framework, StarGAN, the translation model we use, also employs an auxiliary domain classifier D_cls
> to enable translation between multiple domains. Here, D_cls learns to classify images by minimizing the domain classification loss, which is the same as the loss used by the biased classifier of the main task. Hence, if the main classifier is biased, D_cls also tend to learn the same bias instead of the signals.
>
> ---
>
> ### [W2 ColoredMNIST consists of a single source of bias. But how would this method handle multiple types of biases e.g., BiasedMNIST [1]? Does the image translation model uncover all the spurious factors regardless of their ‘ease of exploitation’?]
> Thanks for your insightful and constructive comments. We think that considering multiple biases is a very meaningful direction for future works.
>
> As the reviewer recommended, we conducted the additional experiment on Biased MNIST in the table below. We confirmed that CDvG shows better performance than baselines which use ResNet in [1] when multiple biases exist. Furthermore, we found out that the translation model uncovers ‘digit color’, ‘type of background texture’, ‘background color’, ‘co-occuring letters’ and ‘colors of the co-occuring letters’ to some extent. However, since our method is mainly devised for a single bias factor, CDvG does not translate all these biases perfectly.
>
> **Biased MNIST**
> | **ERM** | **SD** | **Up Wt** | **gDRO** | **PGI** | **Ours** |
> |:-------:|:------:|:---------:|:--------:|:-------:|:--------:|
> |   36.8  |  37.1  |    37.7   |   19.2   |   48.6  |   **49.48**  |
>
> [1] Shrestha, Robik, Kushal Kafle, and Christopher Kanan. "OccamNets: Mitigating Dataset Bias by Favoring Simpler Hypotheses." arXiv preprint arXiv:2204.02426 (2022).
>
> ---
>
> ### [W3 Waterbirds, BAR, ImageNet-9 all test against foreground-background spurious correlations. Apart from completeness, do they add additional value?]
> Since those datasets have different types of background biases, we tried to conduct experiments extensively on various datasets. Furthermore, they have different scales, the number of categories, and the influence of biases:
>
> WaterBird: # of training samples = 4795, 2 categories
>
> BAR: # of training samples =1941, 6 categories
>
> imagenet-9: # of training samples = 54600, 9 categories
>
> ---
>
> ### [W4 It is unclear how the method would tackle real world distributional shifts. Examples, datasets from WILDs benchmark [2] or the different renditions of ImageNet (https://github.com/hendrycks/imagenet-r) could be used, which are not tested here.]
>
> We note that our main target of interest is the highly biased setting, not general cases of distributional shifts. However, we will try our best to do experiments on the referred dataset and will update in the appendix.
>
> ### [W5 The original submission does not include references or the appendix which is referenced in the main text.]
>
> Sorry for the missing reference caused by a mistake in uploading the file.  We have uploaded reference and supplementary materials.
>
> ### [clarity for some sections including Sec. 5.3]
>
> Thank you for your helpful comments. Including Sec 5.3, all parts that lack clarity have been corrected throughout the paper. Note that minor revisions such as simple clarity improvement, grammatical errors, or typo corrections were carried out throughout the paper. We marked in blue the parts that we revised.

---

> > ### Comment · Reviewer_Xnts · 2022-11-22
> > **More concerns**
> >
> > Thank you for responding to the concerns. At this point, I feel like the paper needs more work before acceptance.
> >
> > The following statement raises some concerns:
> >
> > "As we mentioned in Sec. 3.1, we tackle the case where the bias is easier to perceive than the signal, so that the spurious correlation between bias and label is malignant in that the model preferentially takes the bias as a clue to predict the label over signal. Otherwise we do not need a debiasing model."
> >
> > Concern#1: The paper claims that it is "not reliant on any presumptions regarding the biases", which conflicts with the above statement that the method assumes that biases are simpler to exploit. I think the claim needs to be removed.
> >
> > Concern#2: I also disagree with the conclusion that "Otherwise, we do not need a debiasing model". Say, a dataset has varying levels of spurious correlations from simplistic (as used in the paper) to those with similar or greater complexity than the core features. We would still want a bias-resistant model, but it is not clear how the method would behave in this scenario. For instance, since core features are simpler than biases for certain classes, would this approach consider core features themselves to be biased for those classes?
> >
> > Also, as pointed out by the authors, the method does not capture all sources of biases in BiasedMNIST although they are presumably simpler than the core features. I think the paper would benefit from a study on multiple types/levels of biases.
> >
> > Although I am excited about the method, I think the paper needs to better clarify the limitations/potential problems. Specifically, I would encourage the authors to also think in terms of the difficulty of uncovering different types of biases and of scenarios where core features could be mistaken as biases.

---

> > > ### Author Response · Authors · 2022-11-28
> > > **Response to Reviewer Xnts**
> > >
> > > Thank you for the valuable comments. We address the addtional concerns below.
> > >
> > >
> > > ### [Concern#1: The paper claims that it is "not reliant on any presumptions regarding the biases", which conflicts with the above statement that the method assumes that biases are simpler to exploit. I think the claim needs to be removed.]
> > >
> > > Thank you for the suggestion. As we mentioned above, we target to address where bias is malignant (not only ours, but also previous methods such as LfF and DisEnt). However, we fully agree that it can be misleading so we will clarify them.
> > >
> > > ### [Concern#2: I also disagree with the conclusion that "Otherwise, we do not need a debiasing model". Say, a dataset has varying levels of spurious correlations from simplistic (as used in the paper) to those with similar or greater complexity than the core features. We would still want a bias-resistant model, but it is not clear how the method would behave in this scenario. For instance, since core features are simpler than biases for certain classes, would this approach consider core features themselves to be biased for those classes?]
> > >
> > > Thank you for pointing this out. We agree that considering varying levels of spurious correlations from simplistic to those with similar or greater complexity than the core features is meaningful which is a more complex and practical setting. As we mentioned in Sec. 6, it is an important future direction not only for us but also for debiasing community.
> > >
> > > But now, as we mentioned in above response, previous studies including LfF and DisEnt target to address where bias is malignant. Furthermore, we focus on addressing the highly biased setting that bias-free samples are extremely scarce or absent. It is a more malignant problem, therefore a more important issue as Reviewer SxSC and Fqi9 acknowledge, than those targeted in previous studies which tend to break down in the highly biased setting.
> > >
> > > For now, since CDvG (not only ours, but also previous methods such as LfF and DisEnt) focus on handling high-bias situations, their performance on low-bias datasets are limited. In the table below, we conducted the additional experiment that presents the performances of methods under low-bias circumstances. Although not our main focus, they work to some extent (not totally fails). But, they show lower performance than vanilla.
> > >
> > > **CorruptedCIFAR-10**
> > > | **Ratio(%)**       | **Vanilla** | **LfF** | **CDvG** |
> > > |--------------------|:-----------:|:-------:|:--------:|
> > > | 50%                |    **73.92**    |  65.74  |   69.78  |
> > > | 30%                |    **69.98**    |  62.87  |   63.16  |
> > > | 20%                |    **65.89**    |  61.36  |   58.47  |
> > > | 10%                |    **57.40**    |  55.41  |   52.14  |
> > >
> > > Finally we would like to note that we can think of some practical ways to determine whether or not to use debiasing methods when we do not know about the existence of bias in practice. For example, we can train a debiasing model with a given dataset, then in test time, we can slowly adopt the debiasing model; we can maintain two models (vanilla and debiasing models) and use a vanilla as a main classifier and monitor how two models work on unbiased cases in test time. We can adopt it if it shows better performance compared to vanilla, and don't use it otherwise. We also can observe the translated images and judge at least whether the translated factor is a genuine signal or not using our domain knowledge. Therefore, we can decide not to use debiasing methods if the true signal is transformed.
> > >
> > > ### [Also, as pointed out by the authors, the method does not capture all sources of biases in BiasedMNIST although they are presumably simpler than the core features. I think the paper would benefit from a study on multiple types/levels of biases.]
> > >
> > > Sorry for confusion. What we said is that our method captures all sources of biases, but does not perfectly translate all source biases exactly the same as the target biases simultaneously. For example, in some cases, our method translates ‘type of background texture’ into an interpolated version of the original and target while translating all the other biases well. Although our method is mainly devised for a single bias factor, our method showed higher performance compared to the baselines as we showed in the table in the previous response.

---

> > > > ### Comment · Reviewer_Xnts · 2022-12-09
> > > > **Decision**
> > > >
> > > > Thank you for the effort you put in for the rebuttal.
> > > >
> > > > I think this is a neat method, however, even after the edits/response, it still has the same issues:
> > > >
> > > > a) the paper still claims that the method is "not reliant on ANY presumptions regarding the biases"
> > > > b) it lacks a systematic study of types/levels of biases
> > > >
> > > > So, I am unable to recommend a clear acceptance.

---

> > > > > ### Author Response · Authors · 2022-12-09
> > > > > **Response to Reviewer Xnts**
> > > > >
> > > > > Thank you for taking your valuable time to response.
> > > > >
> > > > > First of all, our method targets to address where bias is malignant following the convention of previous SOTAs such as Rebias[1], LfF[2], DisEnt[3], BiaSwap[4], BPA[5] and etc.. Furthermore, we focus on addressing the highly biased setting that bias-free samples are extremely scarce or absent without using the information about biases.
> > > > >
> > > > > a)
> > > > > > To sum it up cleary, we aim to address where bias is malignant and CDvG not reliant on presumptions about the availability of bias-free samples or types of biases.  Therefore, considering varying levels of spurious correlations from simplistic to those with similar or greater complexity than the core features is just out of scope at present (not only ours, but also SOTAs such as Rebias, LfF, DisEnt, BiaSwap, BPA and etc.). As we mentioned before, we totally agree that considering them is meaningful which is a more complex and practical setting and an important future direction for the debiasing community, but now please consider that even the current scope is not solved.
> > > > >
> > > > > b)
> > > > > > Our experiments cover many datasets over different classes (numbers, faces, objects, birds), modalities (images and videos) and biases (synthetic, real) as Revier Fqi9 acknowledge following the convention of SOTAs. Although the final discussion period will soon end, please let us know if you have any further concerns that we have not addressed up to your satisfaction. We will do our best to clarify further.
> > > > >
> > > > >
> > > > > [1] Hyojin Bahng, Sanghyuk Chun, Sangdoo Yun, Jaegul Choo, and Seong Joon Oh. Learning de-biased representations with biased representations. In Hal Daumé III and Aarti Singh (eds.), Proceedings of the 37th International Conference on Machine Learning, volume 119 of Proceedings of Machine Learning Research, pp. 528–539. PMLR, 13–18 Jul 2020.
> > > > >
> > > > > [2] Junhyun Nam, Hyuntak Cha, Sungsoo Ahn, Jaeho Lee, and Jinwoo Shin. Learning from failure: De-biasing classifier from biased classifier. Advances in Neural Information Processing Systems, 33:20673–20684, 2020.
> > > > >
> > > > > [3] Jungsoo Lee, Eungyeup Kim, Juyoung Lee, Jihyeon Lee, and Jaegul Choo. Learning debiased representation via disentangled feature augmentation. Advances in Neural Information Processing Systems, 34:25123–25133, 2021
> > > > >
> > > > > [4] Eungyeup Kim, Jihyeon Lee, and Jaegul Choo. Biaswap: Removing dataset bias with bias-tailored swapping augmentation. In Proceedings of the IEEE/CVF International Conference on Computer Vision, pp. 14992–15001, 2021.
> > > > >
> > > > > [5] Seonguk Seo, Joon-Young Lee, and Bohyung Han. Unsupervised learning of debiased representations with pseudo-attributes. In Proceedings of the IEEE/CVF Conference on Computer Vision and Pattern Recognition, pp. 16742–16751, 2022.

---

### Official Review · Reviewer_SxSC · 2022-10-25

**Confidence:** 3
**Correctness:** 3
**Technical Novelty And Significance:** 3
**Empirical Novelty And Significance:** 3
**Recommendation:** 6

**Clarity, Quality, Novelty And Reproducibility:**

Here are several comments:

1. Where is the reference? It seems the article missing the reference.
2. How you trained StarGAN? Did you train it on Color-MINIST data?
3. How to combine the proposed framework CDvG with LfF? Is the training embedding of CDvG used for LfF?
4. Lacking explanation about StarGAN. Please include a detailed explanation of StarGAN in the supplemental materials.



**Strength And Weaknesses:**

Strength
1. the paper discusses an interesting debiasing case: when there is no unbias data among the data, how can we train an unbias classifier
2. the experiment is very solid

Weakness
1. Detailed experiment setting is not described. For example, how to combine the proposed work with LfF?
2. Missing reference.
3. Missing detailed information about StarGAN, for example, how to train StarGAN on the bias data? What is the structure of StarGAN?


**Summary Of The Paper:**

This paper introduces a contrastive learning-based debiasing framework to train a unbias classifier under the assumption that there are NO unbias samples existing in the training data. To achieve this goal, the authors utilize a conditional StarGAN approach to ``adjust'' the images and augment the samples. The experiment demonstrates that the method can outperform the existing approaches when combing with LfF debiasing framework.

**Summary Of The Review:**

Overall, the paper is well-written and the experiment is solid. But authors should address the comments I mentioned above.

---

> ### Author Response · Authors · 2022-11-17
> **Response to Reviewer SxSC**
>
> We thank the reviewer for the constructive and thoughtful feedback. We address the reviewer’s concerns below.
>
> ### [Where is the reference? It seems the article missing the reference]
>
> Sorry for the missing references caused by a mistake in uploading the file.  We have uploaded the references and supplementary materials.
>
> ### [How you trained StarGAN? Did you train it on Color-MINIST data?]
>
> CDvG trains the StarGAN before training the main classifier on the same given training set as mentioned in the second paragraph of Section 4.2. For example, when the main task is to learn the unbiased classifier with Corrupted-CIFAR10, training a StarGAN on Corrupted-CiFAR10 precedes it.
>
> ### [How to combine the proposed framework CDvG with LfF? Is the training embedding of CDvG used for LfF?]
>
> Sorry for missing a detailed explanation. In the revision, we clarified the details about the combined version in the second paragraph of Sec. 5.2. We marked in blue the parts that we revised.
>
> In the training process, we trained CDvG with concurrently applying LfF’s sample-wise weighting to CE loss for both original images and bias-translated images (LfF proposed sample-wise weighting which gives more weight to bias-free samples and vice versa for bias-aligned samples in cross entropy loss.). Therefore, ‘CDvG+LfF’ allows us to utilize both bias-aligned and bias-free samples to debias by translating bias-aligned samples into bias-free samples (CDvG) and giving more weights to bias-free samples (LfF).
>
> ### [Lacking explanation about StarGAN. Please include a detailed explanation of StarGAN in the supplemental materials.]
>
> Sorry for the confusion. We have uploaded the supplemental material.  In Sec. B, we mentioned detailed information about StarGAN. Please note that we use the default setting of StarGAN.

---

> > ### Comment · Reviewer_SxSC · 2022-12-13
> > **Decision from me**
> >
> > Thank you very much for the response. The response is very helpful. However, after I read other reviewers' comments and the response. I have two main limitations that made me hard to champion this paper.
> >
> > 1) It is hard to justify that the bias introduced in StarGAN is what bias existed in the target data. There are limitations of what types of bias can be alleviated by this framework.
> >
> > 2) since StarGAN is trained in the same training set, it largely reduces the original contribution I would expect. (a pre-trained StarGAN that can be used to debiasing any dataset)
> >
> > In summary, I will not support a clear acceptance of this paper

---

### Official Review · Reviewer_F8oL · 2022-10-25

**Confidence:** 3
**Correctness:** 3
**Technical Novelty And Significance:** 3
**Empirical Novelty And Significance:** 3
**Recommendation:** 5

**Clarity, Quality, Novelty And Reproducibility:**

- The paper should be improved in terms of clarity. The details about the proposed method do not appear until page 6.
- Quality, novelty, and reproducibility look reasonable. The method is simple, and it may be able to be reproduced with the details provided in the paper.

**Strength And Weaknesses:**

Strength:
1. The idea of the paper is simple and can be potentially applied to any dataset. Even if the biases are known, the GAN will supposedly uncover them, so no annotations on biased samples are required.
2. Experiments are conducted on very diverse datasets, including synthetic and real-world images.
3. It is very interesting that the proposed method outperforms previous methods by a large margin when no unbiased samples are available.

Weaknesses
1. Paper clarity should be improved. Details:
- The paper is not very clear about the proposed method. I did not understand what the proposed method was until page 6 (Section 4.2). The paper would benefit from making the proposed model clearer from the beginning.
- It is not clear to me if the image-to-image augmentations are also applied at test time.

2. Looks like the method needs the existence of a very strong bias to be useful. The following points need clarification:
- Why Table 1 only has results for low ratios of unbiased samples? Specially BFFHQ dataset goes up only to 0.5%, while results on other datasets are reported to be up to 5%.
- It should be clarified whether the method works if the dataset bias is not very strong. What if only 50% of the samples are biased? It may be that StarGAN cannot capture this bias well, making the whole method fail. The paper should be clear and transparent about this point.
- In Table 1, except when the ratio is very small, some results do not seem to be statistically significant.

3. Given that this method needs to train a StarGAN model and generate various images per sample in the dataset, computation cost and time should be reported and compared against previous work.

4. Other details:
- Section 5.1 says the model is evaluated on BAR dataset, but results are not reported.
- Why is the Waterbird dataset the only dataset not reported over 3 runs?


**Summary Of The Paper:**

This paper addresses image dataset bias by using GANs and contrastive learning techniques. By training an image-to-image translation model (StarGAN) on a biased dataset, the model learns the implicit biases on that dataset. This is used by the debiasing model to generate new samples with different types of biases. Then, a debiased classifier can be trained using contrastive learning on {original (and biased) sample} - {generated sample} pairs. The model is evaluated on 5 different image datasets and ablation studies are reported.

**Summary Of The Review:**

The proposed method is simple and targets an important task. However, there are some concerns with respect to the limitations that should be clarified during the rebuttal.

---

> ### Author Response · Authors · 2022-11-17
> **Response to Reviewer F8oL (2/2)**
>
> ### [In Table 1, except when the ratio is very small, some results do not seem to be statistically significant.]
>
> First of all, we focus on addressing the highly biased setting that bias-free samples are extremely scarce or absent. It is a more malignant problem, therefore a more important issue as Reviewer SxSC and Fqi9 acknowledge, than those targeted in previous studies which tend to break down in the highly biased setting.
>
> Therefore, by the nature of CDvG, when the ratio of the bias free samples is rather high (that is, 2%, 5% for Corrupted CIFAR-10 and 0.5% for BFFHQ ), it shows comparable or lower performance than LfF which focuses only on bias-free samples. To compensate for this, we further proposed to combine CDvG and LfF which leverage both bias-aligned samples and bias-free samples as mentioned in the second paragraph of Sec. 5.2. ‘CDvG+LfF’ is consistently superior to the baselines with low standard deviation in Table 1. Please note that CDvG can be integrated with other debiasing methods in a plug-and-play manner.
>
> To elaborate on the combined version, we combine CDvG with LfF by training CDvG by concurrently applying LfF’s sample-wise weighting to the cross entropy (CE) loss for both original images and bias-translated images (LfF proposed sample-wise weighting which gives more weight to bias-free samples and vice versa for bias-aligned samples in CE loss.). Please note that we clarified the details about the combined version in the second paragraph of Sec. 5.2 in the revision.
>
> ---
>
> ### [Given that this method needs to train a StarGAN model and generate various images per sample in the dataset, computation cost and time should be reported and compared against previous work]
> CDvG requires additional cost and time to train the translation model and generate translated images before training the main classifier. More precisely, the total number of parameters of StarGAN including a generator and a discriminator is smaller than ResNet-18 which is an auxiliary model of LfF-based models. Thus, computation cost and time for training and inference of each additional model is almost similar, but total training time of the framework of CDvG takes longer time (e.g. 6 hours more on Corrupted CIFAR-10 with a single RTX2080ti) due to sequential training of the translation model and the main classifier whereas LfF-based methods train the auxiliary model in parallel.
>
> ---
>
> ### [Section 5.1 says the model is evaluated on BAR dataset, but results are not reported.]
> We are sorry for the confusion. We decide to omit the results on BAR in Table 1 for the following reason:
> Due to the size of BAR, the baselines initialize the model with a pretrained ResNet.  We found that, unintentionally, the more pretrained weights are kept, the better performances are shown. As a result, when a small learning rate and epochs are used, better performance is obtained. This makes BAR not suitable for evaluating debiasing performance.
> Nevertheless, we reveal that CDvG outperforms baselines on BAR using the same learning rate and epochs specified by LfF. We report the results in the table below.
>
> **BAR**
> | **Vanilla** |  **LfF**  | **DisEnt** |  **CDvG** |
> |-------------|:---------:|:----------:|:---------:|
> | 66.39 ± 0.70 | 63.68 ± 0.49 | 64.84 ± 1.98 | **68.10 ± 0.42** |
>
> ---
>
> ### [Why is the Waterbird dataset the only dataset not reported over 3 runs?]
> To correct, the Waterbird dataset was also reported over 3 runs. For fair comparison, we followed the evaluation process of BPA paper to borrow their reported performance in the paper. To align with the format of BPA, we only presented average performance in Table 3. We clarified this in the caption of Table 3 in the revision.
>
> ---

---

> > ### Comment · Reviewer_F8oL · 2022-12-08
> > **Thank you for the response**
> >
> > Thank you for the clarifications and for updating the paper.
> > Before the rebuttal, I thought an interesting contribution of the paper was the no need to know or assume the type of bias in the dataset. However, after the response, I realized the method does assume that the dataset is highly biased, with very few unbiased samples available. In my option, this is a big assumption, which makes the contribution weaker. Although I can still see merits in the proposed approach, I am leaning toward rejection.

---

> > > ### Author Response · Authors · 2022-12-09
> > > **Response to Reviewer F8oL**
> > >
> > > Thank you for taking your valuable time to response.
> > >
> > > First of all, our method does not assume that the dataset is highly biased.
> > >
> > > Our method targets to address where bias is malignant same as  previous SOTAs such as Rebias[1], LfF[2], DisEnt[3], BiaSwap[4], BPA[5] and etc.. Furthermore, we additionaly focus on addressing the highly biased setting that bias-free samples are extremely scarce or absent. It is a more malignant problem, therefore a more important issue as Reviewer SxSC and Fqi9 acknowledge, than those targeted in previous studies which tend to break down in the highly biased setting. Please note that our method shows better or on par performance on various datasets. (As Reviewer Fqi9 mentioned, our experiments cover many datasets over different classes (numbers, faces, objects, birds), modalities (images and videos) and biases (synthetic, real).
> > >
> > > To the best of our knowledge, there are no studies that focus on bias-aligned samples directly. Also, we showed SOTA performance by combining ours with a bias-free samples based approach, LfF.
> > >
> > > [1] Hyojin Bahng, Sanghyuk Chun, Sangdoo Yun, Jaegul Choo, and Seong Joon Oh. Learning de-biased
> > > representations with biased representations. In Hal Daumé III and Aarti Singh (eds.), Proceedings
> > > of the 37th International Conference on Machine Learning, volume 119 of Proceedings of Machine
> > > Learning Research, pp. 528–539. PMLR, 13–18 Jul 2020.
> > >
> > > [2] Junhyun Nam, Hyuntak Cha, Sungsoo Ahn, Jaeho Lee, and Jinwoo Shin. Learning from failure: De-biasing classifier from biased classifier. Advances in Neural Information Processing Systems, 33:20673–20684, 2020.
> > >
> > > [3] Jungsoo Lee, Eungyeup Kim, Juyoung Lee, Jihyeon Lee, and Jaegul Choo. Learning debiased representation via disentangled feature augmentation. Advances in Neural Information Processing Systems, 34:25123–25133, 2021
> > >
> > > [4] Eungyeup Kim, Jihyeon Lee, and Jaegul Choo. Biaswap: Removing dataset bias with bias-tailored swapping augmentation. In Proceedings of the IEEE/CVF International Conference on Computer Vision, pp. 14992–15001, 2021.
> > >
> > > [5] Seonguk Seo, Joon-Young Lee, and Bohyung Han. Unsupervised learning of debiased representations with pseudo-attributes. In Proceedings of the IEEE/CVF Conference on Computer Vision and Pattern Recognition, pp. 16742–16751, 2022.

---

> ### Author Response · Authors · 2022-11-17
> **Response to Reviewer F8oL (1/2)**
>
> We sincerely appreciate the insightful and constructive comments. We address the reviewer’s concerns below.
>
> ### [Paper clarity should be improved. I did not understand what the proposed method was until page 6 (Section 4.2). The paper would benefit from making the proposed model clearer from the beginning.]
> Thanks for the feedback. Please note that the observation of the image translation models under bias in Sec. 3 is one of our main contributions and a key motivation for understanding our method in Sec. 4 so that it comes before the method. But, to clarify the overall framework, we added a training process of our bias-translation model to the main figure (Fig. 4)  and moved it to page 2 so that the overall flow can be seen at a glance from the beginning in the revision. Also, all parts that lack clarity have been corrected throughout the paper. Note that minor revisions such as simple clarity improvement, grammatical errors, or typo corrections were carried out throughout the paper. We marked in blue the parts that we revised.
> We sincerely appreciate your helpful comments again, which have greatly improved the clarity of the paper.
>
> ---
>
> ### [It is not clear to me if the image-to-image augmentations are also applied at test time.]
> Sorry for the confusion. The image-to-image augmentations are not applied at test time but used only for training unbiased classifiers. We clarified this in the second paragraph of Sec. 4.2 in the revision.
>
> ---
>
> ### [Looks like the method needs the existence of a very strong bias to be useful. The following points need clarification:]
> ### [Why Table 1 only has results for low ratios of unbiased samples? Specially BFFHQ dataset goes up only to 0.5%, while results on other datasets are reported to be up to 5%.]
> For fair comparison with the baselines, we followed the same settings of the baseline papers (LfF, DisEnt, BiaSwap and BPA) for each dataset. (To the best of our knowledge, there are no studies that considered the higher ratios in table 1.)
> Please note that we conduct experiments extensively on various datasets as Reviewer Fqi9 acknowledges.
>
> ---
>
> ### [It should be clarified whether the method works if the dataset bias is not very strong. What if only 50% of the samples are biased? It may be that StarGAN cannot capture this bias well, making the whole method fail. The paper should be clear and transparent about this point.]
> First of all, we focus on addressing the highly biased setting that bias-free samples are extremely scarce or absent. It is a more malignant problem, therefore a more important issue as Reviewer SxSC and Fqi9 acknowledge, than those targeted in previous studies which tend to break down in the highly biased setting.
>
> Nevertheless, we also agree that extending to handling low-bias settings is an important future direction of ours, as we mentioned in Sec. 6. For now, since CDvG (not only ours, but also previous methods such as LfF and DisEnt) focus on handling high-bias situations, their performance on low-bias datasets are limited. In the table below, we conducted the additional experiment that presents the performances of methods under low-bias circumstances. Although not our main focus, they work to some extent (not totally fails). But, they show lower performance than vanilla.
>
> **CorruptedCIFAR-10**
> | **Ratio(%)**       | **Vanilla** | **LfF** | **CDvG** |
> |--------------------|:-----------:|:-------:|:--------:|
> | 50%                |    **73.92**    |  65.74  |   69.78  |
> | 30%                |    **69.98**    |  62.87  |   63.16  |
> | 20%                |    **65.89**    |  61.36  |   58.47  |
> | 10%                |    **57.40**    |  55.41  |   52.14  |
>
> Finally we would like to note that we can think of some practical ways to determine whether or not to use debiasing methods when we do not know about the existence of bias in practice. For example, we can train a debiasing model with a given dataset, then in test time, we can slowly adopt the debiasing model; we can maintain two models (vanilla and debiasing models) and use a vanilla as a main classifier and monitor how two models work on unbiased cases in test time. We can adopt it if it shows better performance compared to vanilla, and don't use it otherwise. We also can observe the translated images and judge at least whether the translated factor is a genuine signal or not using our domain knowledge. Therefore, we can decide not to use debiasing methods if the true signal is transformed.
>
> ---

---

### Official Review · Reviewer_Fqi9 · 2022-10-27

**Confidence:** 5
**Correctness:** 3
**Technical Novelty And Significance:** 2
**Empirical Novelty And Significance:** 3
**Recommendation:** 5

**Clarity, Quality, Novelty And Reproducibility:**

Clarity

(-) Bird-eye view of the proposed framework is missing. Please refer to my summary.

(-) The sentences should be more compact.

(-) The sentences have many redundant descriptions. E.g., "auxiliary information which is rarely obtainable in practice" and "such presumptions about the availability of the auxiliary information or bias-free samples are not always guaranteed"

(-) Sentence flow can be improved. Especially in 3.2 before 3.2.1.

(-) Inconsistent statements:
- "We find that image-to-image translation models favor learning malignant biases over task-relevant signals."
- "We experimentally observe that certain image translation models are also prone to consider conspicuous but incidental information."

(-) It is difficult to guess the expected result in each column in Figure 3. I can see the 2nd and  3rd columns should be female and male. I cannot tell on other datasets.

(-) What do checks and crosses mean in Table 1?

Novelty / Originality

The logical flow is just the same as "Imagenet-pretrained cnns are biased toward texture". Instead of texture, this paper is tackling the biases that can be captured as the domain specific attributes by image-to-image translator.

Quality

This paper is technically sound except that the translators are not always biased.


**Strength And Weaknesses:**

Strengths

(+) The translation model does not require bias supervision for training.

(+) Using translated images with bias for reducing bias of the dataset is plausible.

(+) Experiments cover many datasets over different classes (numbers, faces, objects, birds), modalities (images and videos) and biases (synthetic, real).

(+) The proposed method can be used in any setting.

Weaknesses

(-) The bias of a discriminative model is not guaranteed to be in the same bias of translation model.

(-) Outdated translation model (stargan 2017). Biased translation may not exist in recent translation models: stargan v2 2020, style-aware
discriminator 2022. Is stargan 2017 chosen among other alternatives to produce biases? Please discuss the trade-off between existence
of bias and image quality.

Etc.

How does it work when there is no bias in the dataset?


**Summary Of The Paper:**

This paper tackles the problem of a classifier being biased to nuisance factors (shortcuts). The proposed method uses image-to-image translation to synthesize different biases in the dataset to remove the bias without collecting bias-free dataset.

**Summary Of The Review:**

This paper builds up upon somewhat true (true in some cases) intuition (bias in translation) and provides reasonable de-biasing technique. However, the presentation and the logical flow should be improved. Furthermore, the idea is incremental in that this paper works with image-to-image translation instead of style transfer in "imagenet-pretrained cnns are biased toward texture".

---

> ### Author Response · Authors · 2022-11-17
> **Response to Reviewer Fqi9 (2/2)**
>
> ### [It is difficult to guess the expected result in each column in Figure 3. I can see the 2nd and 3rd columns should be female and male. I cannot tell on other datasets.]
> First column of the images are the original biased images and each of the other columns is the bias-translated images of the leftmost images using our bias-translation model for each class. The type of biases of each dataset are colors, texture, backgrounds and gender, respectively. For example, in Fig.3-(a), the original color-biased digits in the first column are translated to be colored by the corresponding bias color of the target class (e.g red for zero, orange for one, violet for seven). We clarified this in the revision.
>
> ---
>
> ### [What do checks and crosses mean in Table 1?]
> A check means that the model exploits auxiliary information of the bias and a cross means the opposite. Thank you for pointing it out. We clarified this in the caption of Table 1.
>
> ---
>
> ### [The sentences have many redundant descriptions. E.g., "auxiliary information which is rarely obtainable in practice" and "such presumptions about the availability of the auxiliary information or bias-free samples are not always guaranteed"]
> ### [Sentence flow can be improved. Especially in 3.2 before 3.2.1.]
> Sorry for the confusion. Including mentioned redundant descriptions and Sec. 3.2 before 3.2.1,  all parts that lack clarity have been corrected throughout the paper. Note that minor revisions such as simple clarity improvement, grammatical errors, or typo corrections were carried out throughout the paper. We marked in blue the parts that we revised. We sincerely appreciate your thoughtful comments, which have greatly improved the clarity of the paper.
>
> ---
>
> ### [The logical flow is just the same as "Imagenet-pretrained cnns are biased toward texture". Instead of texture, this paper is tackling the biases that can be captured as the domain specific attributes by image-to-image translator.]
> Unlike “Imagenet-pretrained CNNs are biased toward texture" which shows that CNN models tend to be biased to texture than shape, we found that image-to-image translation models tend to learn the bias of each class (which are not just confined to textures) when using biased datasets. Based on this new finding, our key contribution is to do contrastive learning between bias-translated views generated with the biased translation model, which enables learning bias-invariant representations without requiring the presence of bias-free samples inside the dataset.
>
> ---

---

> > ### Comment · Reviewer_Fqi9 · 2022-11-19
> > **Thank you**
> >
> > I appreciate the effort in the revision.
> >
> > > (moved from the first comment to here) Note here that D_cls learns to classify images by minimizing the domain classification loss, which is the same as the loss used by the biased classifier of the main task.
> >
> > I think this argument needs theoretical or empirical support. The biases learned by the main task and the translator may differ.
> >
> > > Image-to-image translation models tend to learn the bias of each class (which are not just confined to textures) when using biased datasets.
> >
> > I am not sure whether this paper introduces generalizable principle or not. It sounds likely but lacks grounding.
> >
> > Alternative way to build a stronger argument would be "We replace multi-task discriminator of the existing translators with discriminator with auxiliary classifier to acquire biased translator. Such replacement ensure that the bias occur in the translator." because StarGAN produces images low quality images limiting the applicability and SOTA translators produces high quality images.

---

> > > ### Author Response · Authors · 2022-11-28
> > > **Response to Reviewer Fqi9**
> > >
> > > Thank you for your constructive comments.
> > >
> > > ### [(moved from the first comment to here) Note here that D_cls learns to classify images by minimizing the domain classification loss, which is the same as the loss used by the biased classifier of the main task. I think this argument needs theoretical or empirical support. The biases learned by the main task and the translator may differ.]
> > >
> > > Since D_cls and the classifier of the main task use almost the same network architecture, the same biased dataset, and the same objective, we think it is natural that D_cls tends to learn the same bias of main classifier. We would appreciate it if you could share why you think the biases may differ.
> > >
> > > Also, as we mentioned in Sec. 3.2.1, we conducted the experiment, which supports the argument empirically, that measures the classification loss on the unbiased dataset with D_cls while training StarGAN on the biased dataset. In Fig. 2, we observed that the accuracy of bias-aligned samples is high, but the accuracy of bias-free samples is low, just like the result of the main task classifier.
> > > In conclusion, high accuracy only on bias-aligned samples means that D_cls also learns the same bias as the main task classifier.
> > >
> > > ### [I am not sure whether this paper introduces generalizable principle or not. It sounds likely but lacks grounding. Alternative way to build a stronger argument would be "We replace multi-task discriminator of the existing translators with discriminator with auxiliary classifier to acquire biased translator. Such replacement ensure that the bias occur in the translator." because StarGAN produces images low quality images limiting the applicability and SOTA translators produces high quality images.]
> > >
> > > Thank you for your insightful and constructive comments. As you suggested, we conducted an experiment in which the multi-task discriminator of the stargan_v2 was replaced with a domain classifier. In the table below, we empirically demonstrate that recent translation models are also inherently bias-susceptible to bias when using a domain classifier. Please note that additional results will be continuously updated due to the large size of the network (6 times larger than stargan) and our computation budget.
> > >
> > > **CorruptedCIFAR-10**
> > > | **Ratio(%)**       | **Vanilla** | **CDvG w/ StarGAN** | **CDvG w/ StarGAN_v2** |
> > > |--------------------|:-----------:|:-------:|:--------:|
> > > | 0.5%                |    20.87    |  31.50  |  **32.30**  |
> > > | 5.0%                |    41.12    |  42.75  |  **47.80**  |
> > >
> > > In conclusion, the use of recent translation models can improve the debiasing performance by improving the quality of translated image, but there is a trade-off between cost and performance. Also, we will clarify the argument about recent translation models which use a multi-task discriminator as you suggested.

---

> > > > ### Comment · Reviewer_Fqi9 · 2022-12-07
> > > > **Thank you. But the concerns remain.**
> > > >
> > > > Thank you for considering my suggestions.
> > > >
> > > > I still have two main concerns.
> > > > 1. The authors claim that the bias of the image-to-image translation models is a new finding. However, StarGAN v2 already states that translating a male to female merely applies make-up without changing other attributes of female.
> > > > 2. There is no guarantee that the auxiliary classifier of the discriminator uses the same bias with the target classifier. For example, for classifying a sample as female, the auxiliary classifier may consider existence of earrings as clue while the target classifier considers lack of gray hair because both can minimize the loss function with the same architecture. This agrees with [W1] and [W2] from reviewer Xnts. As this assumption is the key grounding of this paper, it is a critical reason limiting the generalization capability.

---

> > > > > ### Author Response · Authors · 2022-12-09
> > > > > **Response to Reviewer Fqi9**
> > > > >
> > > > > We sincerely appreciate your effort to make the discussion interactive.
> > > > >
> > > > > 1.
> > > > > > As we mentioned in the second paragraph in Sec. 3.2, CycleGAN presented some typical failure modes that occurs occasionally in 2017. For example, when the source domain is an apple and the target domain is an orange, a transformed image is not the orange counterpart of the input, but an apple with the color and texture of an orange. Considering this phenomenon occurred on an unbiased dataset, we thought that the phenomenon would be exacerbated when handling a biased dataset and therefore examined the behavior of StarGAN to verify this for the first time in Sec. 3.2.1 and Sec 3.2.2.
> > > > >
> > > > > 2.
> > > > > > First of all, our method target to address where bias is malignant same as previous SOTAs such as Rebias[1], LfF[2], DisEnt[3], BiaSwap[4], BPA[5] and etc..
> > > > >
> > > > > > Considering this background, when there is a single malignant bias factor in the biased dataset, both the target classifier and the discriminator use the same malignant bias as a cue for predicition rather than the signal. We support this arguement empirically with the bias-transformed images in Colored MNIST, Corrupted CIFAR-10, BAR and BFFHQ in Fig. 4, accuracy evaluated on bias-aligned and bias-free samples of Colored MNIST, Corrupted CIFAR-10 and BFFHQ using the discrimnator of StarGAN which is near 100% on bias-aligned samples and low on bias-free samples just like the main target classifier in Fig. 2 and final performance improvement compared to SOTAs on various datasets over different classes, modalities and biases in Table 1,  Table 2 and Table 3.
> > > > >
> > > > > > In the case of multiple biases, as Reviewer Xnts recommended, we conducted the additional experiment on Biased MNIST[6] in the table below. We confirmed that CDvG caputures all sources of biases and shows better performance than baselines when multiple biases exist even though our method is mainly devised for a single bias factor. Based on this result, we can infer that the degree of concentration on each bias may slightly differ, but both models will not use different biases as cues.
> > > > >
> > > > >
> > > > > **Biased MNIST**
> > > > > | **ERM** | **SD** | **Up Wt** | **gDRO** | **PGI** | **Ours** |
> > > > > |:-------:|:------:|:---------:|:--------:|:-------:|:--------:|
> > > > > |   36.8  |  37.1  |    37.7   |   19.2   |   48.6  |   **49.48**  |
> > > > >
> > > > >
> > > > > [1] Hyojin Bahng, Sanghyuk Chun, Sangdoo Yun, Jaegul Choo, and Seong Joon Oh. Learning de-biased representations with biased representations. In Hal Daumé III and Aarti Singh (eds.), Proceedings of the 37th International Conference on Machine Learning, volume 119 of Proceedings of Machine Learning Research, pp. 528–539. PMLR, 13–18 Jul 2020.
> > > > >
> > > > > [2] Junhyun Nam, Hyuntak Cha, Sungsoo Ahn, Jaeho Lee, and Jinwoo Shin. Learning from failure: De-biasing classifier from biased classifier. Advances in Neural Information Processing Systems, 33:20673–20684, 2020.
> > > > >
> > > > > [3] Jungsoo Lee, Eungyeup Kim, Juyoung Lee, Jihyeon Lee, and Jaegul Choo. Learning debiased representation via disentangled feature augmentation. Advances in Neural Information Processing Systems, 34:25123–25133, 2021
> > > > >
> > > > > [4] Eungyeup Kim, Jihyeon Lee, and Jaegul Choo. Biaswap: Removing dataset bias with bias-tailored swapping augmentation. In Proceedings of the IEEE/CVF International Conference on Computer Vision, pp. 14992–15001, 2021.
> > > > >
> > > > > [5] Seonguk Seo, Joon-Young Lee, and Bohyung Han. Unsupervised learning of debiased representations with pseudo-attributes. In Proceedings of the IEEE/CVF Conference on Computer Vision and Pattern Recognition, pp. 16742–16751, 2022.
> > > > >
> > > > > [6] Shrestha, Robik, Kushal Kafle, and Christopher Kanan. "OccamNets: Mitigating Dataset Bias by Favoring Simpler Hypotheses." arXiv preprint arXiv:2204.02426 (2022).

---

> ### Author Response · Authors · 2022-11-17
> **Response to Reviewer Fqi9 (1/2)**
>
> We thank the reviewer for the helpful and insightful comments. We address the reviewer’s concerns below.
>
> ### [The bias of a discriminative model is not guaranteed to be in the same bias of the translation model.]
>
> StarGAN, the translation model we use, employs an auxiliary domain classifier D_cls to enable translation between multiple domains. Note here that D_cls learns to classify images by minimizing the domain classification loss, which is the same as the loss used by the biased classifier of the main task. Hence, as the biased classification model does, D_cls also tend to learn the same bias instead of the signals. Please refer to Sec. 3.2.1 and the first paragraph of Sec.3.2.2 .
>
> ---
>
> ### [Outdated translation model (stargan 2017). Biased translation may not exist in recent translation models: stargan v2 2020, style-aware discriminator 2022. Is stargan 2017 chosen among other alternatives to produce biases? Please discuss the trade-off between existence of bias and image quality.]
>
> As mentioned previously, D_cls learns to classify images by minimizing the domain classification loss, which leads the generator to be biased. Thus, any translation model that uses some form of D_cls (including recent models such as stargan v2) would be inherently bias-susceptible, making it compatible with our framework.
> We chose stargan2017 because it is a well-researched and widely-used model that uses a D_cls and fits our computation budget, not because biased translation is a specific artifact of the stargan2017 method. (Please note that the total number of parameters including a generator and a discriminator of stargan2017 is smaller than ResNet18)
>
> ---
>
> ### [How does it work when there is no bias in the dataset?]
> First of all, we focus on addressing the highly biased setting that bias-free samples are extremely scarce or absent. It is a more malignant problem, therefore a more important issue as Reviewer SxSC and Fqi9 acknowledge, than those targeted in previous studies which tend to break down in the highly biased setting.
>
> Nevertheless, we also agree that extending to handling low-bias settings is an important future direction of ours, as we mentioned in Sec. 6. For now, since CDvG (not only ours, but also previous methods such as LfF and DisEnt) focus on handling high-bias situations, their performance on low-bias datasets are limited. In the table below, we conducted the additional experiment that presents the performances of methods under low-bias circumstances. Although not our main focus, they work to some extent (not totally fails). But, they show lower performance than vanilla.
>
> **CorruptedCIFAR-10**
> | **Ratio(%)**       | **Vanilla** | **LfF** | **CDvG** |
> |--------------------|:-----------:|:-------:|:--------:|
> | 50%                |    **73.92**    |  65.74  |   69.78  |
> | 30%                |    **69.98**    |  62.87  |   63.16  |
> | 20%                |    **65.89**    |  61.36  |   58.47  |
> | 10%                |    **57.40**    |  55.41  |   52.14  |
>
> Finally we would like to note that we can think of some practical ways to determine whether or not to use debiasing methods when we do not know about the existence of bias in practice. For example, we can train a debiasing model with a given dataset, then in test time, we can slowly adopt the debiasing model; we can maintain two models (vanilla and debiasing models) and use a vanilla as a main classifier and monitor how two models work on unbiased cases in test time. We can adopt it if it shows better performance compared to vanilla, and don't use it otherwise. We also can observe the translated images and judge at least whether the translated factor is a genuine signal or not using our domain knowledge. Therefore, we can decide not to use debiasing methods if the true signal is transformed.
>
> ---
>
> ### [Bird-eye view of the proposed framework is missing. Please refer to my summary.]
> Thank you for the suggestion. To clarify the overall framework, we added a training process of our bias-translation model to the main figure (Fig. 4) so that the overall flow can be seen at a glance.
>
> ---

---

> > ### Comment · Reviewer_Fqi9 · 2022-11-19
> > **Thank you**
> >
> > Thank you for the effort in the response.
> >
> > > Biased translation may not exist in recent translation models.
> >
> > Recent (2019~) translation methods do not use auxiliary classifier but employ multi-task discriminator.
> >
> > > How does it work when there is no bias in the dataset?
> >
> > This should be discussed in the paper.

---

> > > ### Author Response · Authors · 2022-11-28
> > > **Response to Reviewer Fqi9**
> > >
> > > We sincerely appreciate your time and effort to review our paper.
> > >
> > > ### [Recent (2019~) translation methods do not use auxiliary classifier but employ multi-task discriminator.]
> > >
> > > We address the concern below following your suggestion
> > >
> > > ---
> > >
> > > ### [How does it work when there is no bias in the dataset? This should be discussed in the paper.]
> > > Thank you for your valuable suggestion. We will clarify this in the paper.

---

### Decision · Program_Chairs · 2023-01-20

**Decision:**

Reject

**Justification For Why Not Higher Score:**

Reviewers and the AC found the approach to be novel and interesting, but were unconvinced about the generality of the method and felt that the present paper is limited by strong assumptions. The paper could be improved either by weakening the assumptions, showing more thoroughly that they are valid assumptions that hold in real scenarios, or better articulating the limitations of the method in the writing.

**Justification For Why Not Lower Score:**

N/A

**Metareview: Summary, Strengths And Weaknesses:**

Summary:
This paper presents a method to debias image datasets. First, image-to-image translation is used to create multiple views of the data each with a different bias. Then, this multiview data is fed to a contrasative learner that learns a representation invariant to these biases. The method works without having access to labeled biases, instead using the natural biases that emerge during translation, which are assumed (and empirically verified) to overlap with the biases of the downstream classifier.

Strengths:
* Interesting to use the natural biases of the domain translator as a source of supervision for debiasing
* Good results across a range of datasets

Weaknesses:
* There are some strong assumptions that may limit the generality and real world use of the method: 1) assumption that the discriminative model and the translation model share the same biases, 2) assumption that the biases are easier to learn than the core features.
* Performance is weak when the bias is not strong (e.g., when only 50% of samples are biased)
* StarGAN is somewhat out of date and it's not clear if the same findings would hold for more recent translation approaches (the initial experiments on StarGANv2 are a good step)



**Summary Of Ac-Reviewer Meeting:**

This paper was discussed over email between the AC and reviewers. While the reviewers felt that the rebuttal and revisions had improved the paper (especially it's clarity) they still felt it did not quite meet the bar for ICLR, raising the weaknesses described above. The one positive reviewer agreed with the weaknesses raised by the others and a consensus formed to reject the paper.